# Fluorometric and Colorimetric Biosensors for the Assay of Cholinesterase Inhibitors

**DOI:** 10.3390/s25092674

**Published:** 2025-04-23

**Authors:** Miroslav Pohanka

**Affiliations:** Military Faculty of Medicine, University of Defense, Trebesska 1575, 500 02 Hradec Kralove, Czech Republic; miroslav.pohanka@unob.cz or miroslav.pohanka@gmail.com

**Keywords:** acetylcholinesterase, assay, butyrylcholinesterase, enzyme, gold nanoparticles, nerve agents, neurotoxin, photogrammetry, peroxidase

## Abstract

Cholinesterases, specifically acetylcholinesterase (AChE) and butyrylcholinesterase (BChE), play critical roles in neurotransmission and are key targets for inhibitors with therapeutic and toxicological significance. This review focuses on the development and application of fluorometric and colorimetric biosensors for the detection of cholinesterase inhibitors. These biosensors take advantage of the unique properties of AChE and BChE to provide sensitive and selective detection methods essential for environmental monitoring, food safety, and clinical diagnostics. Recent advances in assay techniques, including the use of gold nanoparticles, pseudoperoxidase nanomaterials, and innovative enzyme–substrate interactions, are highlighted. This review also discusses challenges and future directions for optimizing these biosensors for practical applications, emphasizing their potential to enhance public health and safety.

## 1. Introduction

Acetylcholinesterase (AChE) and butyrylcholinesterase (BChE) are two known cholinesterases. AChE is a crucial enzyme in the nervous system, playing a significant role in neurotransmission, in which it is primarily responsible for breaking down the neurotransmitter acetylcholine at synaptic junctions, ensuring the termination of nerve impulses and preventing the continuous stimulation of muscles, which is vital for normal muscle function and cognitive processes [1]. BChE, while less specific in its substrate preference, acts as a backup to AChE and is involved in the hydrolysis of various choline-based esters. It also plays a role in drug metabolism, detoxification, and the less-specific hydrolysis of various esters [2,3]. Together, these enzymes maintain the balance of neurotransmitter levels, contributing to the proper functioning of the nervous system and general homeostasis.

Compounds that inhibit AChE and butyrylcholinesterase BChE are known as cholinesterase inhibitors and have significant therapeutic and toxicological implications [4]. These inhibitors, such as donepezil, rivastigmine, and galantamine, are commonly used in the treatment of Alzheimer’s disease to enhance cholinergic transmission by preventing the breakdown of acetylcholine, thus improving cognitive function [5]. However, other inhibitors, such as organophosphates and carbamates, found in certain pesticides and nerve agents, can lead to the excessive accumulation of acetylcholine, resulting in prolonged muscle contraction, respiratory failure, and potentially fatal outcomes [6,7,8,9]. The balance between therapeutic benefits and toxic risks underscores the importance of understanding and regulating these compounds.

In analytical chemistry, AChE and BChE are often employed as biorecognition elements in biosensors designed to detect inhibitors [10,11]. These enzymes interact specifically with analytes that inhibit their activity, allowing for the quantification of such compounds. For instance, in the detection of pesticides or nerve agents, AChE and BChE-based biosensors can measure the degree of enzyme inhibition, which correlates with the concentration of the inhibitor present [12,13,14]. This interaction is crucial for environmental monitoring, food safety, and clinical diagnostics, providing a sensitive and selective method for detecting harmful substances. The use of these enzymes in biosensors highlights their importance in ensuring public health and safety.

The aim of this review paper focused on fluorometric and colorimetric assays involving AChE and BChE is to provide a comprehensive and critical survey of recent advancements in these analytical techniques. Though the cholinesterase biosensors have already been reviewed by many authors, this research introduces new applications and approaches, as new materials and technologies are evolving. Therefore, this review focuses on new trends and surveys actual research in the field of colorimetric and fluorometric cholinesterase biosensors.

Using the current literature, this review seeks to highlight the development and optimization of assays that leverage the unique properties of AChE and BChE as biorecognition elements for the detection of inhibitors. It will examine the sensitivity, specificity, and practical applications of these assays in various fields, such as environmental monitoring, food safety, and clinical diagnostics. Additionally, the review identifies emerging trends, challenges, and future directions in the use of fluorometric and colorimetric methods, offering valuable insights for researchers and practitioners in the field.

## 2. Common Principles in Fluorometric and Colorimetric Biosensors

Fluorometric methods in analytical chemistry involve measuring the intensity of fluorescent light emitted via a substance when it is exposed to ultraviolet or visible light. These methods are highly sensitive and can detect low concentrations of analytes. Fluorometry is often used in biochemical and medical applications, such as detecting specific proteins or nucleic acids. The technique relies on the principle that certain compounds absorb light at one wavelength and emit it at a longer wavelength. The path source of light to the sample may have an angle (typically 90°) to the path between the sample and the light detector. Biosensors using fluorometry can contain biorecognition elements providing fluorescence on a thin surface, or the fluorescent product can be released to the ambient medium. By measuring the emitted light, scientists can quantify the concentration of the fluorescent compound in a sample. The common principle of a fluorometric biosensor is depicted in Figure 1.

Colorimetric methods, on the other hand, involve measuring the intensity of color produced via a chemical reaction. These methods are based on the principle that the concentration of a substance can be determined by the color intensity of the solution. Colorimetric analysis is widely used in various fields, including environmental monitoring, clinical diagnostics, and food quality control. Colorimetric biosensors evolve from the standard colorimetric methods by attaching biorecognition elements on a solid surface; a chromogenic compound can be presented in the ambient medium or attached to the surface along with the biorecognition element. The colored product can apply color to a sheet with a thin layer of a biosensor or to the medium. The intensity of the color is then measured using a colorimeter or spectrophotometer, allowing for the quantification of the analyte. The path of light can go through the medium or be reflected backwards, depending on the geometry between the source of light and the light detector. The scheme of a colorimetric analysis is shown in Figure 2.

Fluorometric and colorimetric biosensors are innovative tools used to detect and quantify biological molecules. Fluorometric biosensors utilize fluorescent signals to indicate the presence of target analytes, offering high sensitivity and specificity. These sensors are particularly useful in applications where detecting low concentrations of substances is crucial, such as in medical diagnostics and environmental monitoring. Colorimetric biosensors rely on visible color changes to signal the presence of target molecules. They are valued for their simplicity and ease of use, making them suitable for point-of-care testing and field applications. Both types of biosensors play an important role in advancing analytical techniques and improving the accuracy and efficiency of biological assays. Some recent papers on the topic can be cited as examples. An analytical method based on a fluorometric biosensor was, for example, developed by Li and colleagues [15]. The biosensor used chitosan-grafted Fe,Cu-doped carbon dots (CDs), which are used to reduce KMnO_4_ to produce a composite compound, CS@Fe,Cu/CDs-MnO_2_. This compound exhibited high catalytic activity and a fluorescence quantum yield of 76%. The biosensor operated in colorimetric and fluorometric modes. In the colorimetric mode, the CS@Fe,Cu/CDs-MnO_2_ acted like a peroxidase enzyme, oxidizing 3,3′-diaminobenzidine tetrahydrochloride and H_2_O_2_ to produce a brown product of 3,3′-diaminobenzidine tetrahydrochloride. When endotoxins were present, they interacted with the biosensor, reducing its catalytic activity and altering the colorimetric response. In fluorometric mode, the oxidized 3,3′-diaminobenzidine tetrahydrochloride product quenched the fluorescence of CS@Fe,Cu/CDs-MnO_2_ through an inner-filter effect and static quenching. The presence of endotoxins affected the fluorescence intensity, providing a measurable signal. The biosensor performance was enhanced by using a smartphone to analyze the results. It showed a linear response to endotoxin concentrations in specific ranges with low limits of detection (LOD) for both colorimetric and fluorometric methods. The sensor was also tested with real injection samples, demonstrating reliable recovery rates and low relative standard deviation and indicating its effectiveness and reliability. Reports of other promising applications were recently published, like descriptions of biosensors an on assay of γ-aminobutyric acid [16], ethanol [4], uric acid [17], amygdalin [18], and human immunodeficiency virus [19].

Colorimetric and fluorometric methods are alternatives to electrochemistry and to instrumental methods such as chromatography. Every approach offers its advantages and disadvantages, and no method can be taken as universally better than another. Optical and electrochemical methods are the two important platforms for the construction of biosensors, and the issue was surveyed, for example, with biosensors for an antioxidant assay in food [20], the detection of *Escherichia coli* [21], and biosensors based on engineered peptides [22]. Colorimetric and fluorometric biosensors have become increasingly applicable due to the spread of cheap cameras, colorimetric laboratory instruments, and smart devices with various optical sensors. The cheap optical devices used for colorimetry can involve some limitations, including inconsistency in quality, aberrations in imaging, a limited dynamic range, and low sensitivity. On the other hand, colorimetric biosensors offer some inherent advantages, such as possible control of the signal via the naked eye when the coloration is visible and the use of disposable material such as cuvettes or assay cells from inexpensive polymers. These advantages contrast with electrochemical methods, for which, typically, expensive raw material like noble metals is required, and visual control is not possible. The matrix effect and interferences can be an issue in all platforms, the interference of the matrix effect must be considered, and there is no common advantage or disadvantage. While some, for instance, redox-active or other interfering compounds can cause a problem in an electrochemical assay, other compounds like a colored one or precipitated cloud can prevent results from being achieved using optical assays. Fluorometric biosensors involve higher demands of technical hardware than simple colorimetric biosensors because simple cameras are typically not sensitive enough for standard fluorometric assays. The higher price of a fluorometric apparatus is balanced with higher sensitivity and improvements in other analytical specifications.

## 3. Cholinesterases 

AChE (EC 3.1.1.7) is a crucial enzyme primarily found at neuromuscular junctions and cholinergic brain synapses, and is also bound to erythrocytes. AChE can be bound by glycosylphosphatidylinositol, which anchors cell membranes [23]. Structurally, AChE is a serine hydrolase with a catalytic triad composed of serine, histidine and glutamate residues, and it is close to the alpha anionic site (also known as the catalytic anionic site), which is responsible for interaction with quaternary ammonium of acetylcholine during hydrolysis, and the substrate gains proper orientation during the reaction [24]. The human type of AChE contains amino acids’ residues in the positions Ser-203, His-447, and Glu-334 [25]. The catalytic triad is deep inside the enzyme and it is connected via the protein surface through an active-site gorge and outer entrance through a peripheral anionic site [26,27]. Its primary physiological role is to terminate synaptic transmission by hydrolyzing the neurotransmitter acetylcholine into choline and acetate, thus ensuring that nerve impulses are not continuously propagated. This rapid breakdown of acetylcholine is essential for muscle function, cognitive processes, and overall regulation of the nervous system.

BChE (EC 3.1.1.8), also known as pseudocholinesterase or plasma cholinesterase, is an enzyme that shares structural similarities with AChE but differs in its substrate specificity and distribution. The active site of BChE has some similarities to AChE, and the mechanism of substrate hydrolysis is also based on the same principle, a less-developed alpha anionic site, an active-site gorge, and a peripheral anionic site, which are the major structural differences [28,29,30]. BChE is predominantly found in the liver, plasma, and the central nervous system. Compared to AChE, which can be found only at specific sites, BChE occurs freely, and it remains in the plasma or serum when blood is processed [31,32,33]. It hydrolyzes butyrylcholine more efficiently than acetylcholine and plays a secondary role in cholinergic neurotransmission. It should be observed that BChE is named after butyrylcholine, just as AChE is named after the neurotransmitter acetylcholine. However, butyrylcholine is an artificial substrate of BChE, but it does not naturally occur in the human body. BChE is involved in the metabolism of various ester-based drugs and the detoxification of certain xenobiotics, contributing to the defense mechanisms against toxic compounds. The hydrolysis of cocaine can be cited as an example [34,35,36].

## 4. Inhibitors of AChE and BChE

Inhibitors of cholinesterases have significant therapeutic and toxicological implications. Drugs such as donepezil, rivastigmine, and galantamine are used in the treatment of Alzheimer’s disease by inhibiting AChE, thus increasing acetylcholine levels and improving cognitive function. Natural compounds such as huperzine A also exhibit cholinesterase inhibitory activity. However, certain pesticides, such as carbofuran, and nerve agents such as sarin, irreversibly inhibit cholinesterases, leading to the accumulation of acetylcholine and causing severe neurological symptoms or death [37,38,39]. The dual nature of cholinesterase inhibitors highlights their importance in both medical therapy and toxicology.

The inhibitors do not exert unique mechanisms in how they act on enzymes, and not all inhibitors have the same potency for both cholinesterases. While some can easily inhibit both AChE and BChE, others are selective of only one cholinesterase. The mechanism of how an inhibitor is bound to the cholinesterase should be considered when an assay with cholinesterase as a recognition part is developed. A large group of organophosphate nerve agents such as sarin or soman and former pesticides such as paraoxon and malaoxon are potent inhibitors of AChE and BChE. Oganophosphorus compounds cause the irreversible inhibition of the binding of AChE and BChE to the serine of a catalytic triad [40]. There are some exceptions of organophosphate inhibitors that are selective of one of the cholinesterases, but the number of them is quite low. The compound called iso-OMPA or tetraisopropylpyrophosphamide is a known selective inhibitor of BChE [41].

Carbamate inhibitors such as carbofuran pesticide and the drugs rivastigmine or pyridostigmine are bound to the same site as organophosphates (catalytic triad serum), and most of them exert an equal affinity for AChE and BChE [42]. While the organophosphorus residuum of an organophosphate in the active site of cholinesterase is stable, and the inhibition cannot be spontaneously reverted, the inhibition via a carbamate is reverted after some time. The inhibition is called pseudoirreversible, as the carbamoyl residuum bound to the serine of the active site undergoes spontaneous hydrolysis, resulting in the return of cholinesterase activity [43,44].

There are several reversible binding compounds that bind to AChE and BChE and cause the inhibition of their activity. It can be generalized that most compounds reversibly inhibiting cholinesterases interact with aromatic amino acids of an alpha anionic site, an active-site gorge, and a peripheral anionic site. Because, for AChE, the active-site gorge and peripheral anionic site are more developed and contain more aromatic amino acids, most of the reversible inhibitors exert a higher affinity for AChE than BChE, and BChE is either less or nearly not inhibited by them. Noncompetitive inhibition is the most common type of inhibition that the compounds exert. The drugs donepezil and tacrine, and natural compounds such as caffeine, can be cited as examples of noncompetitive inhibitors of AChE with significantly lower or no affinity to BChE and with the ability to bind to an alpha anionic site, an active-site gorge, and a peripheral anionic site [45,46]. Huperzine A drug inhibits AChE with mixed mechanisms in which noncompetitive and competitive inhibition is involved [47,48], and the drug galantamine causes the competitive inhibition of AChE because it binds to its alpha anionic site [49,50]. The general information on the main types of cholinesterase inhibitors mentioned in this chapter is presented in Table 1.

## 5. Cholinesterase Activity Assays and Cholinesterase Biosensors

Cholinesterase biosensors are analytical devices that can detect the presence of compounds that inhibit AChE and BChE. They are not able to simply distinguish between specific inhibitors. The cholinesterase biosensors work on the principle of measuring changes in cholinesterase activity, and the reduction in activity is proportional to the concentration of the inhibitor. These biosensors typically exert high sensitivity and low limits of detection, depending on the type of inhibitor analyzed and the fact of whether the inhibitor exerts a reversible or irreversible mechanism of action. The irreversible inhibitors bind covalently to the active site and remain bound during the assay. This is an advantage when a highly diluted sample should be analyzed but the volume of the sample is not a problem. These compounds can also covalently react with other structures chemically resembling active sites; this reaction is not relevant for the assay, though it can hypothetically worsen the limit of detection. During an assay, the biosensor can be placed to sample and interact with enzymes with contaminated liquid or air contaminated with vapors of the irreversible inhibitor, such as a nerve agent, which gradually causes inhibition until all immobilized enzymes are without activity. It is obvious that the limit of detection is closely related to the time of incubation with a sample in these cases. Shortening the time for which the biosensor is exposed to the sample with an irreversible inhibitor causes some enzymes to remain active, and the decrease in enzyme activity is not so significant. On the contrary, when time is not a limiting factor, the biosensor can be placed in the contaminated sample even until the complete eradication of enzyme activity in the biosensor. The sensitivity of the cholinesterase biosensors is further enhanced via the effect of signal multiplication. Enzyme activity is measured through biochemical procedures working on the principle that cholinesterase converts an enzyme substrate, which eventually yields a product that can be measured via electrochemistry, optics, or other approaches. Thousands of such substrate molecules are converted into products by one molecule of an enzyme. A single molecule of an inhibitor causes thousands of such products not to form. Cholinesterases also have quite high turnover rates for the common substrates; this specification further enhances the sensitivity of the assay. The common principle of cholinesterase biosensors is shown in Figure 3.

An assay of reversible inhibitors using a cholinesterase biosensor is less common than an irreversible one. Typical cholinesterase biosensors are devices that can detect any of the inhibitors that have an affinity for the enzyme immobilized in the biosensor. The practical relevance relates to military nerve agents and pesticides because these compounds represent a major risk. Reversible inhibitors typically do not represent a substantial risk that leads to the need to analyze them in the field or under other conditions where a biosensor is more suitable than standard analytical procedures. Reversible inhibitors can be washed out from the enzyme, which is why false negativity can occur because of an assay. The rate of reversible inhibitor dissociation from an active site can be slow, and it differs from inhibitor to inhibitor. However, both reversible and irreversible inhibitors can be detected with cholinesterase biosensors, and when an assay is focused on the detection of nerve agents or pesticides that inhibit cholinesterases, the possibility of interference via the reversible inhibitors should be considered.

Cholinesterase biosensors may use various chemical reactions for their proper function. Some common reactions are mentioned in this chapter, and the less common reactions developed for a particular device are mentioned in the next chapter.

Several applications centered on the principle of electrochemistry. It should be noted that these biosensors provided quite good results. The substrate acetylcholine hydrolyzed to thiocholine and acetic acid via AChE bound to a biosensor followed by the oxidation of thiocholine in a voltametric assay served as a platform in several articles [51,52,53,54]. Similarly, biosensors with BChE, in which the hydrolysis of butyrylthiocholine to thiocholine and butyric acid was also developed, followed by voltametric oxidation [55]. Other electrochemical biosensors use the hydrolysis of acetylcholine, which is converted into acetic acid and choline, and the choline is subsequently oxidized via choline oxidase to betaine and hydrogen peroxide, as a coproduct in the reaction is measured using voltammetry [56,57,58,59]. A variant with BChE, butyrylcholine and cholineoxidase is also known [60].

Methods for measuring cholinesterase activity can also be based on optics, and many protocols have been published on this topic; some assays are quite common, even in clinical practice, where the activity of AChE and BChE is taken for biochemical markers [61]. The Ellman assay is probably the most common method for determining the activity of AChE and BChE. The assay was developed as a rapid colorimetric determination of AChE activity in the early 1960s by George L. Ellman and coworkers, and a corresponding report was published in 1961 [62]. The assay was used in various studies in which AChE activity was measured in clinical samples, as can be learned from the cited papers [63,64,65]. There are also variants of the Ellman assay for determining BChE activity [66,67]. The principle behind the Ellman assay to measure AChE activity, and it involves the enzymatic hydrolysis of acetylthiocholine, a synthetic substrate, via cholinesterase enzymes (Figure 4). This reaction produces thiocholine and acetate. Thiocholine then reacts with 5,5′-dithiobis-(2-nitrobenzoic acid), commonly known as Ellman’s reagent. This reaction results in the formation of a yellow product, 5-thio-2-nitrobenzoic acid, which can be quantified by measuring its absorbance at 412 nm using a spectrophotometer. The intensity of the yellow color is directly proportional to the amount of thiocholine produced and, thus, to the cholinesterase activity in the sample. When the absorbance values are converted to a standard curve, the enzyme activity can be accurately determined. The variant of the Ellman assay for the measurement of BChE activity involves the use of butyrylthiocholine instead of acetylthiocholine. Although the Ellman assay is common in standard spectrophotometry, it is also found to be used in the construction of biosensors, where many applications are based on immobilized cholinesterases and their reagents in drop form on their surface [68,69,70].

There are also other methods that provide colored compounds through catalyzed conversion using cholinesterases. Indoxylacetate, providing blue indigo due to hydrolysis catalyzed via BChE or with a lower turnover rate for AChE, can be used as an example [71,72,73]. A fluorogenic and chromogenic ester such as 2,6-dichloro-indophenyl acetate [74] or derivatives of resorufin [75,76,77] can serve as an example of other substrates for the enzyme activity assay. The recent applications of colorimetric and fluorometric assays in cholinesterase biosensors are discussed in the next chapter. There can be a difference between the standard optical assay and the biosensor assay with respect to the application of a substrate for the cholinesterase activity assay. While the standard optical test is based on the enzyme and the substrate in homogeneous phases, the biosensor contains immobilized cholinesterase, the substrate can simply be adherent to the enzyme nearby and become solved during the assay, and there can be a chromogenic reagent attached along with the enzyme and the signal, or the substrate can be added to the sample. Specific applications are mentioned in more detail.

## 6. Recently Developed Colorimetric and Fluorometric Cholinesterase Biosensors

The recent development of colorimetric and fluorometric biosensors is related to advanced devices with quite intriguing sensitivity combined with the possibility of applying them in the point-of-care field or other similar situations. There have also been notable reviews in which colorimetric and fluorometric cholinesterase biosensors are discussed [78,79,80,81,82,83,84,85,86,87,88,89].

In specific examples, Peng and coworkers developed a colorimetric biosensor suitable for the detection of pesticide carbaryl using a green photocatalytic biosensor [90]. This biosensor operates on the principle of inhibiting AChE activity. The biosensor uses a double-strand DNA-SYBR Green I complex to oxidize 3,3′,5,5′-tetramethylbenzidine into a blue-colored oxidized form of 3,3′,5,5′-tetramethylbenzidine. AChE catalyzes the production of thiocholine, which reduces oxidized 3,3′,5,5′-tetramethylbenzidine to its colorless form. However, the presence of carbaryl inhibits AChE activity, resulting in less thiocholine production and a higher concentration of oxidized 3,3′,5,5′-tetramethylbenzidine, which can be measured colorimetrically. The biosensor demonstrates high sensitivity with a limit of detection as low as 0.008 ng/mL and a linear response range from 0.01 to 0.25 ng/mL. It also shows good selectivity against non-target chemicals and can effectively detect carbaryl in real water samples.

A high-precision colorimetric-fluorescent dual-mode biosensor designed for detecting AChE as a marker was performed using a trimetallic nanozyme (Cu-Fe-ZIF-8/P-GCNN) with peroxidase-mimicking properties [91]. The principle of the biosensor is based on the inhibition of AChE activity, which affects the oxidation of 3,3′,5,5′-tetramethylbenzidine and prevents the colorimetric reaction. AChE catalyzed the hydrolysis of acetylcholine to produce thiocholine, which reduces the oxidized 3,3′,5,5′-tetramethylbenzidine back to its colorless form. When AChE activity is reduced in the sample due to the presence of inhibitors, such as certain pesticides or neurotoxins, the reduction in 3,3′,5,5′-tetramethylbenzidine is hindered, resulting in a measurable colorimetric and fluorescent signal. The biosensor demonstrates high sensitivity with a limit of detection of 0.13 μU/mL for colorimetric detection and 0.04 μU/mL for fluorescence detection, and it shows good selectivity and stability for a practical AChE assay. This biosensor is not intended to be used for an assay of an inhibitor, and AChE itself is an analyte. Nevertheless, it is mentioned here because the analytical platform can be easily adapted for an inhibitor assay and the colorimetric-fluorescent principle of the assay is quite an interesting approach.

A colorimetric biosensor was designed for the detection of organophosphate pesticides using AChE and silver nanoparticles immobilized on an alginate–chitosan film matrix [92]. The principle of this biosensor is based on the inhibition of AChE activity via organophosphates that were represented as pesticide profenofos. AChE hydrolyzes acetylthiocholine to produce thiocholine, which induces the aggregation of silver nanoparticles, resulting in a color change from brownish yellow to pale yellow. The presence of profenofos causes the inhibition of AChE, reducing thiocholine production and thus preventing a color change. The biosensor’s response is measured by analyzing the red–green–blue (RGB) values of the film. This biosensor demonstrates high sensitivity with a limit of detection of 0.04 mg/L and a linear detection range from 0.05 to 6.00 mg/L. It offers rapid, in situ, and real-time detection without the need for sophisticated instruments, making it suitable for monitoring organophosphate levels in food and environmental samples.

A colorimetric biosensor for detecting organophosphate pesticides, specifically parathion ethyl, using AChE and cysteamine-capped gold nanoparticles as a nanozyme, was described in a paper by Shah et al. [93]. The principle of this biosensor is based on the inhibition of AChE via organophosphates, followed by a reduction in acetylthiocholine and choline production. In the absence of organophosphates, AChE hydrolyzes acetylcholine, producing choline, which induces the aggregation of cysteamine-capped gold nanoparticles, thereby reducing their catalytic activity. The cysteamine-capped gold nanoparticles catalyze the oxidation of 3,3′,5,5′-tetramethylbenzidine to a blue-colored product, which can be measured colorimetrically. The presence of parathion ethyl led to the inhibition of AChE, resulting in less choline production and maintaining the catalytic activity of cysteamine-capped gold nanoparticles, which leads to a stronger colorimetric signal. Though the parathion is not a strong inhibitor of AChE, its degradation product paraoxon is a compound that covalently binds to the active site of AChE. The biosensor demonstrates high sensitivity with a limit of detection of 5.8 ng/mL and a linear detection range from 11.6 to 92.8 ng/mL.

A paper by Lu et al. discusses a dual-mode colorimetric–photothermal biosensor designed for detecting AChE activity and its inhibitors, such as paraoxon ethyl [94]. The biosensor utilizes an Fe–N–C nanozyme with peroxidase-like activity to catalyze the oxidation of 3,3′,5,5′-tetramethylbenzidine in the presence of hydrogen peroxide, producing a blue-colored product. The principle of the assay is based on the inhibition of the Fe–N–C nanozyme’s activity by thiocholine, which is generated from acetylthiocholine in the presence of AChE. The presence of AChE leads to the production of thiocholine, which blocks the active sites of the Fe–N–C nanozyme, reducing its catalytic activity and resulting in a decrease in the colorimetric and photothermal signals. The biosensor demonstrates high sensitivity with limits of detection of 1.9 mU/mL (colorimetric) and 2.2 mU/mL (photothermal) for AChE, and 0.012 μg/mL (colorimetric) and 0.013 μg/mL (photothermal) for paraoxon ethyl. This dual-mode detection platform offers a simple, cost-effective, and portable method for monitoring AChE activity and its inhibitors in various samples.

A single-atom rhodium nanozyme used for the highly sensitive colorimetric detection of AChE activity and adrenaline was developed by Guan et al. [95]. This biosensor leverages the oxidase-like activity of the rhodium nanozyme to catalyze the oxidation of 3,3′,5,5′-tetramethylbenzidine into a blue-colored product. The principle of the assay is based on the inhibition of the rhodium nanozyme’s activity via thiocholine, which is produced from acetylthiocholine in the presence of AChE. The presence of AChE leads to the production of thiocholine, which inhibits the rhodium nanozyme, reducing the colorimetric signal. The biosensor demonstrates high sensitivity with a limit of detection of 3.2 × 10^−4^ mU/mL for AChE and 0.106 μg/mL for adrenaline, making it suitable for detecting these compounds in various samples. This biosensor does not serve for assay of an inhibitor but AChE. Nevertheless, the approach is quite interesting and potentially applicable to the next use of an inhibitor assay.

A smartphone-assisted colorimetric biosensor for detecting organophosphorus pesticides represented by the insecticide dipterex on the peel of fruits was described in another paper [96]. This biosensor utilizes PtCu_3_-alloy nanocrystals with peroxidase-like activity to catalyze the oxidation of 3,3′,5,5′-tetramethylbenzidine into a blue-colored product. The principle of the assay is based on the inhibition of AChE using dipterex. AChE hydrolyzes acetylthiocholine to produce thiocholine, which inhibits the oxidation of 3,3′,5,5′-tetramethylbenzidine, resulting in a faded color. In the presence of dipterex, AChE activity is inhibited, leading to less thiocholine production and a restored blue color. The biosensor demonstrates high sensitivity with a limit of detection of 0.05 mU/mL for AChE and 0.5 ng/mL for dipterex. The concentrations of organophosphorus pesticides can be determined by comparing the B/RG value (brightness value of blue divided by those of red and green) of a test strip with a calibration curve using a smartphone. This method offers a simple, cost-effective, and portable solution for monitoring organophosphorus pesticide residues in fruits, contributing to food safety.

A colorimetric biosensor using AuHg aerogels for the sensitive detection of AChE activity and organophosphorus pesticides was published by Wu et al. [97]. The biosensor leverages the peroxidase-like activity of AuHg aerogels, which is significantly enhanced through the incorporation of mercury onto gold nanowires. This enhancement allows the biosensor to operate effectively in a nearly neutral pH environment, overcoming the acidic pH limitation of traditional Au-based nanozymes. The principle of the assay is based on the inhibition of the peroxidase-like activity of AuHg aerogels through thiocholine, a product of the AChE-catalyzed hydrolysis of acetylthiocholine. In the presence of organophosphates, AChE activity is inhibited, leading to less thiocholine production and a restored peroxidase-like activity, resulting in a measurable color change. The biosensor demonstrates high sensitivity with a limit of detection of 0.02 mU/mL for AChE and 0.86 ng/mL for paraoxon ethyl as a model organophosphate.

In a paper by Cha and coworkers, there was a simple and sensitive colorimetric biosensor for detecting organophosphorus pesticides using AChE encapsulated in naturally occurring extracellular vesicles [98]. The principle of the biosensor is based on the inhibition of AChE activity by organophosphates. AChE catalyzes the hydrolysis of acetylthiocholine to produce thiocholine, which reacts with 5,5′-dithiobis (2-nitrobenzoic acid) to form a yellow-colored product, 5-thio-2-nitrobenzoic acid. In the presence of an organophosphate, AChE activity is inhibited, resulting in a reduced colorimetric response. The biosensor demonstrates high sensitivity with a limit of detection of 53.8 pmol/L for paraoxon ethyl, a representative of organophosphates. The use of extracellular vesicles provides a protective environment for AChE, enhancing its stability and making the biosensor suitable for detecting organophosphates in aqueous solutions and spiked human serum samples.

Considering recent studies on colorimetric and fluorometric biosensors with cholinesterases, these findings can be summarized into the main trends. While there are two cholinesterases, AChE is the first choice for biosensor development, and BChE is out of interest. Unfortunately, the authors do not discuss their choice in cited papers, and the reasons can only be estimated. The better availability of AChE, the higher turnover rate for standard substrates, and the higher sensitivity to most of the relevant inhibitors are the reasons. In-depth comparisons of biosensors with AChE and BChE have not been made. Probably some benefits of BChE remain overlooked because this enzyme can provide quite good stability, and the reduced anionic sites in the enzyme provide lower sensitivity to weak reversible inhibitors, so these compounds can interfere in an assay at a lower rate. The authors also chose common inhibitors, such as paraoxon. They are good for a mutual comparison of biosensors. However, these compounds have specific chemical and physical specifications. An assay optimized for them can be less effective for compounds such as nerve agents that are typically more volatile, and some of them are chemically less stable. All the recent papers exerted decent analytical specifications, and when the limits of detection are considered, they can compete with the more expensive instrumental laboratory methods. The papers devoted to colorimetric and fluorometric biosensors are summarized in Table 2.

## 7. Future Trends

Considering recent studies on colorimetric and fluorometric biosensors with cholinesterases, these findings can be summarized as several main trends. While there are two cholinesterases, AChE is the first choice for biosensor development, and BChE is not of interest. Unfortunately, the authors do not discuss their choice in the cited papers, and their reasons can only be estimated. The better availability of AChE, the higher turnover rate for standard substrates, and the higher sensitivity to most of the relevant inhibitors are the reasons. In-depth comparisons of biosensors with AChE and BChE have not been carried out. Some benefits of BChE probably remain overlooked because this enzyme can provide quite good stability and the reduced anionic sites in the enzyme provide lower sensitivity to weak reversible inhibitors, so these compounds can interfere in an assay at a lower rate. The authors also chose common inhibitors, such as paraoxon. It is good for the mutual comparison of biosensors. However, these compounds have specific chemical and physical specifications. An assay optimized for them can be less effective for compounds such as nerve agents, which are typically more volatile, and some of them chemically less stable. All of the recent papers presented decent analytical specifications, and considering the limits of detection, they can compete with laboratory methods that use more expensive instruments.

Despite significant advancements in the development of fluorometric and colorimetric biosensors for cholinesterase inhibitors, several gaps remain in the current literature. Most studies focus on AChE as the primary enzyme for biosensor development, with butyrylcholinesterase BChE receiving less attention. Comparative studies evaluating the performance of biosensors based on AChE versus BChE are scarce, yet such studies could reveal potential advantages of BChE, such as its stability and lower sensitivity to weak reversible inhibitors. While many biosensors demonstrate high sensitivity, their specificity and selectivity towards different inhibitors, especially in complex matrices, need further improvement. The interference from non-target compounds and the matrix effect can significantly affect the accuracy of these biosensors. Additionally, the stability of the enzymes used in biosensors is a critical issue, as enzyme degradation can lead to false results, necessitating more robust and reproducible biosensors that can maintain their functionality over extended periods. Although some biosensors are designed for field applications, their practical usability in real-world scenarios, such as environmental monitoring and food safety, is often limited by factors like the need for sophisticated instruments or complex sample preparation procedures. The cost of biosensor production, including the manufacturing of pure enzymes, remains high, highlighting the need for developing cost-effective production methods without compromising the biosensor’s performance. To address these gaps, future research should focus on conducting comprehensive studies comparing the performance of biosensors based on AChE and BChE; developing strategies to improve the specificity and selectivity of biosensors; enhancing the stability of the enzymes used in biosensors through enzyme engineering, immobilization techniques, or the development of synthetic enzyme mimics; designing and developing biosensors that are truly field-ready with minimal requirements for additional equipment or complex sample preparation; exploring innovative methods for the cost-effective production of biosensors; and encouraging interdisciplinary research that combines expertise from fields such as materials science, biochemistry, and engineering. By addressing these gaps and focusing on these key areas, future research can significantly advance the field of fluorometric and colorimetric biosensors for cholinesterase inhibitors, ultimately enhancing their practical applications and impact on public health and safety.

## 8. Conclusions

This review underscores the significant progress made in the development of fluorometric and colorimetric biosensors for cholinesterase inhibitors. These biosensors have demonstrated high sensitivity and specificity, making them invaluable tools in various fields such as environmental monitoring, food safety, and clinical diagnostics. The integration of advanced materials like gold nanoparticles has enhanced the performance of these biosensors, enabling the detection of low concentrations of inhibitors. Despite these advancements, challenges remain, including the need for improved stability, reproducibility, and cost-effectiveness of the biosensors. Because an enzyme is used in these biosensors, researchers should consider possible malfunctions caused by its degradation or partial degradation, resulting in false analysis results. Manufacturing a pure enzyme is also a procedure involving demands of equipment, raw materials, and personnel. The development of simple but reliable protocols for cholinesterase manufacturing is a step preceding the production of a practical device based on a cholinesterase biosensor.

Future research should focus on addressing these challenges and exploring new materials and methods to further enhance the capabilities of cholinesterase biosensors. The continued development and optimization of these biosensors hold great promise for improving public health and safety by providing the reliable and efficient detection of harmful substances.

## Figures and Tables

**Figure 1 sensors-25-02674-f001:**
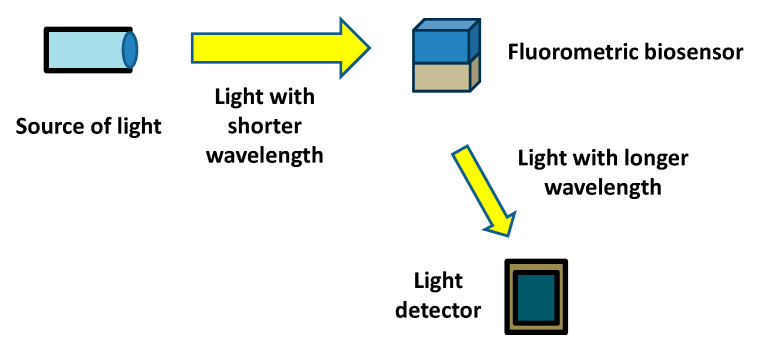
Common principle of a fluorometric biosensor with main parts necessary for its proper function highlighted.

**Figure 2 sensors-25-02674-f002:**
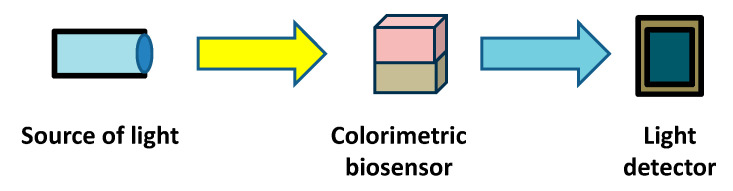
Common principle of a colorimetric biosensor with main parts necessary for its proper function highlighted.

**Figure 3 sensors-25-02674-f003:**
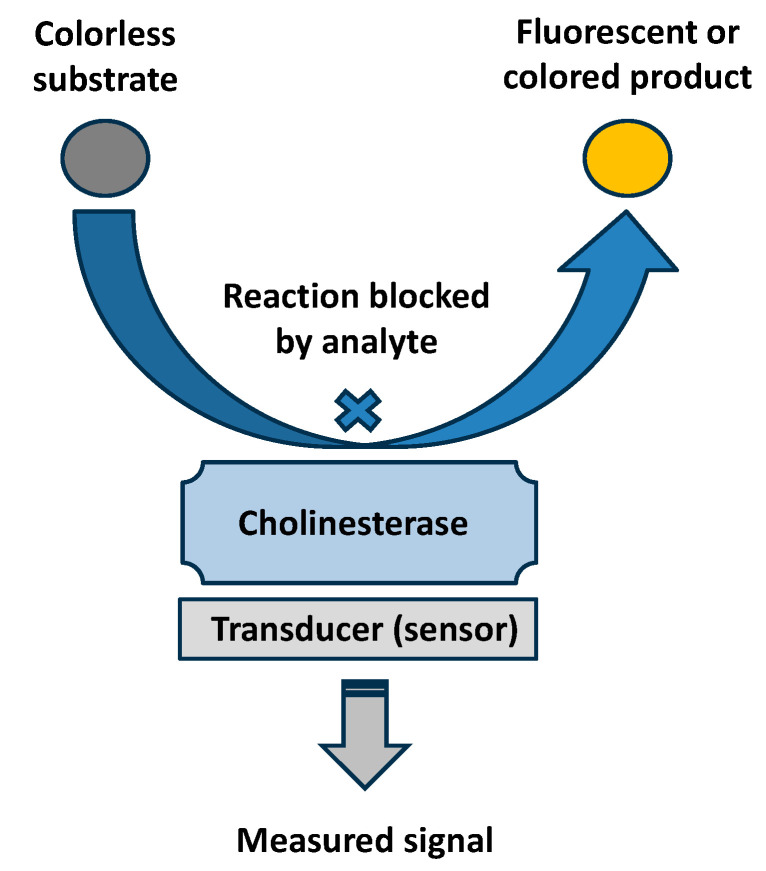
Common principle of a colorimetric or fluorometric cholinesterase biosensor for an assay of an enzyme inhibitor.

**Figure 4 sensors-25-02674-f004:**
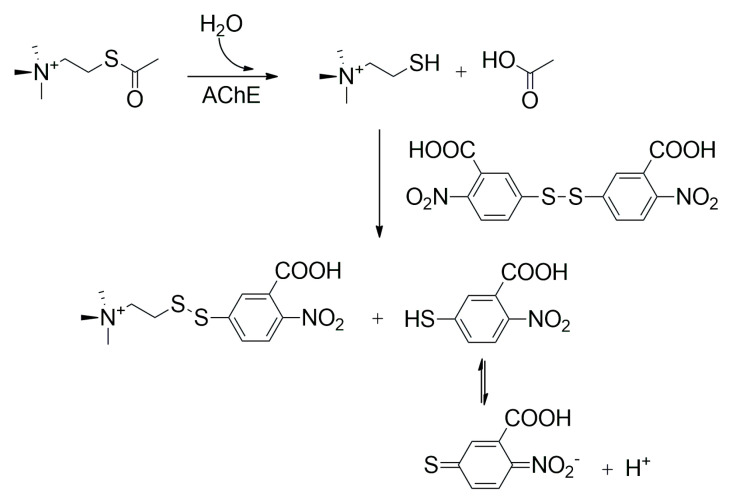
A common principle of the Ellman assay for AChE activity.

**Table 1 sensors-25-02674-t001:** Survey of cholinesterase inhibitors and their specifications.

Type of Inhibitor	Chemical Compounds and Examples	Site of Action	Targeted Cholinesterase
Irreversible	Organophosphate nerve agents like sarin or soman and former pesticides like paraoxon and malaoxon	Serine of catalytic triad	AChE and BChE
Pseudoirreversible	Carbamate inhibitors like pesticide carbofuran and drugs rivastigmine or pyridostigmine	Serine of catalytic triad	AChE and BChE
Noncompetitive	The drugs donepezil and tacrine, as well as natural compounds like caffeine	Alpha anionic site, active-site gorge, and peripheral anionic site	Mainly AChE
Mixed noncompetitive and competitive	Huperzine A	Alpha anionic site, active-site gorge, and peripheral anionic site	AChE
Competitive	Galantamine	Mainly alpha anionic site	AChE

**Table 2 sensors-25-02674-t002:** Survey of recently developed biosensors.

Type of Biosensor	Analyte	Analytical Specifications	References
AChE biosensor based on conversion of acetylthiocholine to thiocholine followed by multiple steps with including redox reaction of 3,3′,5,5′-tetramethylbenzidine	Carbaryl	Limit of detection: 0.008 ng/mL; linear response: 0.01 to 0.25 ng/mL	[90]
Colorimetric-fluorescent dual-mode biosensor with a trimetallic nanozyme with peroxidase-mimicking properties	AChE	Limit of detection: 0.13 μU/mL for colorimetric detection and 0.04 μU/mL for fluorescence detection	[91]
A colorimetric biosensor with AChE and silver nanoparticles immobilized on an alginate–chitosan film	Profenofos	Limit of detection of 0.04 mg/L and a linear detection range from 0.05 to 6.00 mg/L	[92]
A colorimetric biosensor with AChE and cysteamine-capped gold nanoparticles as a nanozyme	Parathion ethyl	Limit of detection of 5.8 ng/mL and a linear detection range from 11.6 to 92.8 ng/mL	[93]
Colorimetric-photothermal biosensor containing Fe–N–C nanozyme with peroxidase-like activity	Paraoxon ethyl, AChE	Limits of detection of 1.9 mU/mL (colorimetric) and 2.2 mU/mL (photothermal) for AChE, and 0.012 μg/mL (colorimetric) and 0.013 μg/mL (photothermal) for Paraoxon ethyl	[94]
Rhodium nanozyme for highly sensitive colorimetric detection of AChE activity	AChE	limit of detection of 3.2 × 10^−4^ mU/mL for AChE	[95]
Smartphone-assisted colorimetric biosensor utilizing PtCu_3_ alloy nanocrystals with peroxidase-like activity	AChE and pesticide dipterex	Limit of detection of 0.05 mU/mL for AChE and 0.5 ng/mL for dipterex	[96]
A colorimetric biosensor using AuHg aerogels exerting peroxidase-like activity	AChE and paraoxon ethyl	0.02 mU/mL for AChE and 0.86 ng/mL for paraoxon ethyl	[97]
A colorimetric biosensor using AChE encapsulated in naturally occurring extracellular vesicles	Paraoxon ethyl	Limit of detection of 53.8 pmol/L	[98]

## Data Availability

All data are presented in this paper.

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
