# Peer review of "Fluorometric and Colorimetric Biosensors for the Assay of Cholinesterase Inhibitors"

_sensors, 2025, doi:10.3390/s25092674_

Round 1
Reviewer 1 Report
Comments and Suggestions for Authors
The review “Fluorometric and Colorimetric Biosensors for the Assay of Cholinesterase Inhibitors” describes recent developments in fluorometric and colorimetric biosensors for the detection of cholinesterase inhibitors. Various sensors are presented, including those based on gold nanoparticles, pseudoperoxidase nanomaterials, and others. The author discusses the challenges and future directions of sensor development for these analytes, including practical applications. Overall, the work is written in clear language and is well-structured. However, there are several areas that require improvement, as outlined below:
- Compare colorimetric and fluorescent methods for the detection of cholinesterase inhibitors with electrochemical, chromatographic and other methods in Section 2.
- The section titled “Recently Developed Colorimetric and Fluorometric Cholinesterase Biosensors” reads somewhat like a literature review. There is a lack of critical evaluation and data analysis. Which sensors are the most sensitive among those mentioned? Which signal fragments and receptors are primarily used?
- There are two sections in the article that are both numbered 5.
- In conclusion, give specific examples of ways to improve the stability, reproducibility and cost-effectiveness of biosensors.
Author Response
I appreciate the time and effort you devote to providing detailed comments and recommendations. Your insights have helped us to address several critical points, refine our arguments, and improve the overall coherence of the manuscript.
Specifically, I have made the following revisions based on your feedback.
- Thank you for the suggestion. The comparison of colorimetric and fluorometric methods is given to Section 2.
- The discussion in Chapter 5 was improved. Please, see new paragraphs.
- The number of sections was corrected.
- The conclusion chapter was improved, and new text was added there.
Reviewer 2 Report
Comments and Suggestions for Authors
The review contains numerous well known general information (e.g. the fluorescence/adsorption measurement) as for the rest is just a continuous summarization of some papers (Team A presents method X, team B published method Y, team Z did other variation) without a clear evaluation of the advantages and disadvantages, the real benefits of the method (if any), nor a comparison between techniques. The limitation of the methods (colored solutions), interferences and the fact that these methods are not applied in commercial laboratories are not considered. Perspectives and remaining issues are absent.
“Biosensors” by definition imposed by the Biosensors and Bioelectronics journal implies the biological component is immobilized and not free in the solution. Fig1 and 2, Ellman and the other reagents from page 7-8 imply free enzyme. The title indicate that it is about cholinesterase inhibitors, but other analytes are included: assay γ-aminobutyric acid [16], ethanol [4], uric acid [17], amygdalin [18], and human immunodeficiency virus [19]. In fact, the entire section 2. Fluorometric and colorimetric methods is too general and should be removed.
Caption is wrong: ” Table 1. This is a table. Tables should be placed in the main text near to the first time they are cited.”
202The irreversible inhibitors bound covalently to the active site and remain bound during the assay. // Why only to the active site and not other locations?
Figure 3. is too general and not relevant.
232 Reversible inhibitors can be washed out from the enzyme, which is why false negativity can occur because of an assay. // the reversibility reaction rate for cholinesterase is not that fast and numerous biosensors successfully detect carbamates or other compounds. Also, the organophosphates are reversible initially, the enzyme is completely destroyed only after “aging”.
For reference 85 it is not indicated the inhibitor.
331 The presence of parathion ethyl inhibits AChE// parathion is not a strong inhibitor, paraoxon is the inhibitor
Author Response
I appreciate the time and effort you devote to providing detailed comments and recommendations. Your insights have helped us to address several critical points, refine our arguments, and improve the overall coherence of the manuscript.
Specifically, I have made the following revisions based on your feedback.
- The common discussion in the review was improved including giving expected trends and the reasons why it was written.
- The section was improved and discussion was added to better fit the common theory to the manuscript.
- The caption of Table 1 was corrected.
- The discussion of irreversible inhibitors was improved in the manuscript.
- Figure 3. was re-drawn to better fit the manuscript.
- The discussion of reversible inhibitors was improved. Ad organophosphates: to the best of my knowledge, these compounds are irreversible inhibitors from the beginning. The aging process (spontaneous de-alkylation) changes some chemical specifications, but it does not have an effect on the activity. The enzyme is fully inhibited before and afterward. The aging process has some relevance for therapy. After aging, oxime reactivators such as obidoxime cannot be used. Nevertheless, this fact is not an issue in this review, which is focused on analytical methods.
- In reference 85 (now 91), no inhibitor used. The authors used AChE as an analyte. I mentioned this study because of the assay protocol. I understand that this may be confusing to the readers. I improved the text about this study and wrote why the citation given to this manuscript.
- I thank to reviewer for comment on parathion ethyl. It is true. I modified the text.
Reviewer 3 Report
Comments and Suggestions for Authors
This review aims to provide an up-to-date analysis of fluorometric- and colorimetric-based biosensors for detecting AChE and BChE, highlighting their applications in environmental monitoring, food safety, and clinical diagnostics. While the authors effectively present the advantages of these sensing platforms, they fail to offer a meaningful comparison with other technologies, such as electrochemical and FET-based sensors. Moreover, the review lacks a clear emphasis on research gaps in comparison to previously published work and does not explicitly state its purpose, limiting its relevance to the sensing and biosensing community in its current form. Additionally, several key points require further clarification and elaboration, as outlined below.
1) The Table 1 caption has to be rewritten
``This is a table. Tables should be placed in the main text near to the first time they are cited``
2) Authors should clearly emphasize the specific gap their review paper aims to address and explicitly state the primary purpose of the study, especially given the existence of similar works. Doing so will better highlight the manuscript's significance and its unique contribution to the field.
3) Improving the quality of the figures is highly recommended. Additionally, incorporating illustrative images from previously published work could help better depict the detection mechanism and various functionalization stages of biosensors.
4) The manuscript lacks the perspective visions, in which authors should provide pertinent suggestions and insights on improving the existing sensing platform or addressing commercialization challenges. Accordingly, authors are suggested to discuss the barriers of translating biosensors from lab to market.
5) The English should be carefully reviewed, as many sentences require refinement. The authors are also advised to avoid using long sentences.
Author Response
I appreciate the time and effort you devote to providing detailed comments and recommendations. Your insights have helped us to address several critical points, refine our arguments, and improve the overall coherence of the manuscript.
Specifically, I have made the following revisions based on your feedback.
- The common discussion in the review was improved, including giving expected trends and the reasons why it was written.
- The table caption was rewritten.
- The aims of the manuscript are better defined now.
- Figures 1-3 were redrawn.
- Visions were added to the manuscript and the merit of cholinesterase biosensors is better discussed.
- The whole manuscript was checked.
Round 2
Reviewer 1 Report
Comments and Suggestions for Authors
Dear Author,
I am satisfied with your response to my comments.
Author Response
Thank you for your positive recommendation.

Reviewer 2 Report
Comments and Suggestions for Authors
Only one minor modification "Though the parathion is not a strong inhibitor of AChE, its reaction product paraoxon is
a compound covalently binding to active site of AChE." Replace "reaction " by degradation or oxidation
Author Response
Thank you for your positive recommendation. The sentence about paraoxon was corrected.

Reviewer 3 Report
Comments and Suggestions for Authors
I have no further comments. The manuscript is scientifically sound and can be accepted for publication in its current form.
Author Response

(The authors gave the same response as above.)
